# Development of a Prognostic Nomogram for Liver Metastasis of Uveal Melanoma Patients Selected by Liver MRI

**DOI:** 10.3390/cancers11060863

**Published:** 2019-06-21

**Authors:** Pascale Mariani, Sylvain Dureau, Alexia Savignoni, Livia Lumbroso-Le Rouic, Christine Levy-Gabriel, Sophie Piperno-Neumann, Manuel J. Rodrigues, Laurence Desjardins, Nathalie Cassoux, Vincent Servois

**Affiliations:** 1Department of Surgical Oncology, Institut Curie, 26 rue d’Ulm, 75248 Cedex 05 Paris, France; livia.lumbroso@curie.fr (L.L.-L.R.); christine.levy@curie.fr (C.L.-G.); laurence.desjardins@curie.fr (L.D.); nathalie.cassoux@curie.fr (N.C.); 2Department of Biostatistic, Institut Curie, 26 rue d’Ulm, 75248 Cedex 05 Paris, France; sylvain.dureau@curie.fr (S.D.); alexia.savignoni@curie.fr (A.S.); 3Department of Medical Oncology, Institut Curie, 26 rue d’Ulm, 75248 Cedex 05 Paris, France; sophie.piperno-neumann@curie.fr (S.P.-N.); manuel.rodrigues@curie.fr (M.J.R.); 4Department of Radiology, Institut Curie, 26 rue d’Ulm, 75248 Cedex 05 Paris, France; vincent.servois@curie.fr

**Keywords:** uveal melanoma, liver metastasis, prognostic factors, liver MRI, tumour burden

## Abstract

Patients with liver metastases of uveal melanoma (LMUM) die from their metastatic evolution within 2 years. We established a nomogram to choose a treatment adapted to life expectancy. From 2002 to 2013, we reviewed 224 patients with LMUM selected by liver MRI. A nomogram was developed based on a Cox model. The predictive performance of the model was assessed according to the C-statistic, Kaplan–Meier curve, and calibration plots. The median follow-up was 49.2 months (range, 0.6–70.9). The survival rates at 6, 12, and 24 months were 0.88 (0.95 CI [0.84–0.93]), 0.68 (0.95 CI [0.62–0.75]), and 0.26 (0.95 CI [0.21–0.33]), respectively. The four factors selected for the nomogram with a worse prognosis were: A disease-free interval between the UM and LMUM groups of less than 6 months (HR = 3.39; 0.95 CI [1.90–6.05]), more than 10 LMUM (HR = 3.95; 0.95 CI [1.97–4.43]), a maximum LMUM of more than 1200 mm2 (HR = 2.47; 0.95 CI [1.53–3.98]), and a lactate dehydrogenase (LDH) value greater than 1.5 (HR = 3.72; 0.95 CI [2.30–6.00]). The model achieved relatively good discrimination and calibration (C-statistic 0.71). This nomogram could be useful for decision-making and risk stratification for therapeutic options.

## 1. Introduction

Among the patients treated for uveal melanoma (UM), 50% will develop metastatic disease during follow-up when they have risk factors [1]. These risk factors are based on clinical characteristics of UM (largest basal diameter, maximum tumor thickness) and on genomic analysis of the primary tumor [2,3]. In this setting, a monosomy of chromosome 3 and chromosome 8q gain (M3/8g) has been reported to be a prominent risk factor for metastases [4]. When metastatic disease occurs, it is exclusively hepatic in more than 90% of patients. The majority of these patients will die from this metastatic evolution within 2 years [5,6].

For those patients who do develop metastatic disease, there is no proven standard of care so far. Chemotherapeutic regimens and various combinations have been investigated in UM with disappointing results to date [7]. In contrast to the metastases of cutaneous melanoma, immunotherapy—approved in the USA and Europe for treatment of advanced melanoma—gives poor results in metastatic UM [8]. Given the limited activity of agents that are currently approved for advanced melanoma in the treatment of metastatic uveal melanoma, clinical trials using targeted therapies are being conducted to optimize patient management of care. Pending the results of these clinical trials, the choice of treatment remains difficult. Classically, for patients who have a high tumor burden in the liver or extrahepatic metastases, a systemic treatment is proposed. For patients who are oligometastatic, we can also propose invasive therapeutic means. This can be surgical resection of the liver metastases [9,10], coupled or not to radiofrequency ablation [11], hepatic intra-arterial chemosaturation [12,13,14], or other intra-arterial therapies [15,16]. These locoregional approaches allow a transient control of the metastases, but nearly all patients relapse. Nevertheless, there is a population of long-term survivors, who have slowly progressive metastases and for which any treatment results in a stable disease. It is currently not clear whether this is due to the action of the proposed treatment or a naturally quiescent disease [17,18].

Because clinicians lack a standard prognostic tool, it seems useful to propose a reproducible prognostic algorithm, based on the analysis of retrospective data, that could be integrated in clinical practice. Only one study of this type has been published to date [19]. This study involved 254 patients recruited from two centers. However, the data collected for these patients ranged from 1990 to 2013 for the primary center, which can induce considerable bias concerning the validity of the imaging data. In particular, liver magnetic resonance imaging (MRI) has experienced important technological developments during this period. According to our experience, MRI is an imaging method for objectively and retrospectively assessing the importance of hepatic involvement in UM, like in other types of cancer [20,21,22].

Several retrospective studies have assessed the prognostic factors in patients with liver metastases of uveal melanoma (LMUM) [23,24,25,26,27,28]. The main independent prognostic factors common to these different studies include clinical and biological factors, as well as the evaluation of liver tumor burden determined by different imaging methods. In three more recent retrospective studies [19,29,30], the previously described prognostic factors appear as the main factors determining patient survival: age, performance status, delay in metastasis (DFI), LDH level [19,30] or alkaline phosphatases [29], and the extent of hepatic invasion assessed on imaging. Hepatic invasion can be assessed either by measuring the largest diameter of the largest metastasis [29,30] or by measuring the percentage of liver invasion [19].

Among the previously mentioned prognostic factors, the measurement of the importance of hepatic invasion with imaging is the most variable and the least codified. The results may depend on the imaging method used (US, CT, MRI) and the method of measuring tumor invasion (number of metastases evaluated, one-dimensional (1D), two-dimensional (2D), or three-dimensional (3D) measurement using either geometric models or a precise manual segmentation of the lesions. The international rules for the evaluation of tumors in 1D (RECIST 1.1) [31] or in 2D (World Health Organization—WHO) [32] are well established. The overall assessment of tumor burden by 3D methods remains to be established. Given currently available data for metastatic uveal melanoma, the imaging mode used and the method of assessing tumor burden on imaging remains to be standardized. From this point of view, MRI has appeared for many years an effective imaging method for evaluating the spread of liver metastatic disease in several types of cancer [21,22]. For many years we have been performing liver MRI in some of our patients suspected of metastatic evolution of their primary ocular tumor. We could therefore retrospectively analyze the MRI data archived in our center since 2002.

The objective of our study was to establish a nomogram on a mono-centric cohort of patients with liver metastases of uveal melanoma (LMUM), whose hepatic invasion was retrospectively analyzable on liver MRI performed at the time of the initial diagnosis of metastatic disease in order to be integrated in clinical practice.

## 2. Results

Two hundred and twenty-four patients were treated at the Institut Curie for LMUM and underwent hepatic MRI in our institution at the onset of their metastatic disease, the results of which are archived on the Picture Archiving Communication System (PACS) and retrospectively analyzable. The median follow-up of the population was 49.2 months (range, 0.6–70.9 months). The median age of the patients was 57 years (IQR, 49–66). Metastases were exclusively located in the liver in 90% of patients. Metastases were synchronous to the diagnosis of UM in 35% of patients. The PS status was 0–1 in 95% of patients. The median overall survival after metastasis was 1.34 years [range, 0–5.9 years]. The survival rates at 6, 12, and 24 months were, respectively, 0.88 (0.95 CI [0.84–0.93]), 0.68 (0.95 CI [0.62–0.75]), and 0.26 (0.95 CI [0.21–0.33]).

### 2.1. Univariate Analysis

The results of univariate analysis are shown in Table 1. Significant variables were the age at diagnosis of UM, ciliary body involvement, type of UM treatment, time to onset of metastasis after UM diagnosis, type of LMUM treatment, extrahepatic metastatic involvement, PS status, and the LDH value, as well as the following observed on MRI:localization of LMUM, number of liver segments with LMUM, number of LMUM, surface area of the largest LMUM, and presence or absence of miliary disease.

### 2.2. Multivariate Analysis

Multivariate analysis identified four independent prognostic factors of survival (Table 2). The disease-free interval between UM and LMUM survival was decreased when the disease-free interval between UM and LMUM was short (HR = 3.39, 0.95 CI [1.90–6.05] for a period of 0 to 6 months (*p*-value <0.001); and HR = 2.02, 0.95 CI [1.24–3.27] for a period of 6 to 12 months). For the number of LMUM, survival was decreased when a higher number of lesions was detected (HR = 1.58, 0.95 CI [1.07–2.33] for 5 to 10 lesions; and HR = 2.95, 0.95 CI [1.97–4.43] for more than 10 lesions, *p*-value <0.001). For the size of the largest LMUM, survival was decreased when the surface area exceeded 800 mm2 (HR = 1.74, 0.95 CI [1.00–3.05] for a surface area between 801 and 1200 mm2; and HR = 2.47, 0.95 CI [1.53–3.98] for a surface area more than 1200 mm2, *p*-value <0.001). For the LDH value, survival was decreased when LDH was above 1.5 times the normal value (HR = 3.72, 0.95 CI [2.30–6.00], *p*-value <0.001). Therefore, these 4 variables were used to build the nomogram.

### 2.3. Prognostic Profile Nomogram

A prognostic nomogram for overall survival with scales for the above four factors was constructed (Figure 1). The bars above the different classes of the variables correspond to the confidence intervals. In terms of discrimination, the average Harrell C-Index value on 400 bootstrap samples was 0.71 (0.95 CI). The model well predicts the condition of the patients in 71% of cases. Calibration of the model was evaluated with the calibration curves of the model at the three time points used in the nomogram: 6 months, 12 months, and 24 months (Figure 2). The calibration of our nomogram was better at 24 months than at 12 and 6 months, and better at 12 months than at 6 months. At 6 months, the model tended to overestimate the survival of patients. At 12 months, the model was well calibrated for patients with a predicted survival ranging from 0.5 to 0.9. Below a survival of 0.5, the corrected calibration curve of our model moves away from the diagonal. The model underestimated survival, with a gradual increase in this underestimation as the survival decreased. At 24 months, the calibrated curve of our nomogram remained very close to the diagonal. Therefore, the survival predicted by our model was very close to the observed survival.

## 3. Discussion

The objective of our study was to establish a nomogram to assist in decision-making regarding the choice of treatment for patients with LMUM, selected by liver MRI. The multivariate analysis identified four variables as adjusted prognostic factors for survival after metastasis—time to onset of metastasis, number of LMUM on MRI, surface area of the LMUM on MRI, and LDH value at the diagnosis of LMUM.

Our results confirmed almost exclusive hepatic involvement when the patient became metastatic (90% of cases). Concerning the clinical characteristics of UM and its treatment, none of the significant variables selected in the univariate analysis were significant in the multivariate analysis. Notably, age and ciliary body involvement are not retained in the final model because the information they convey and their effect on survival is at best minimal after adjusting for all variables. One of the hypotheses is that the ciliary body involvement, which is a known prognostic factor for the onset of metastases, has no effect once the patient is metastatic. It is possible that the characteristics of the initial tumor do not affect survival as soon as metastases have appeared, or that their effect is embedded within the DFI variable if they have contributed to metastases occurring faster. The presence of extrahepatic metastases associated with hepatic metastases was not prognostic in the multivariate analysis, probably because of its rarity. The time to the onset of metastasis was significant in multivariate analysis. The other 3 significant variables reflected the importance of hepatic invasion: LDH> 1.5 N, number of LMUM, and surface area of the largest LMUM. LDH levels and tumor burden are well-established prognostic markers in different types of cancer (lymphoma, cutaneous melanoma), although the interdependence of these two parameters is currently still being discussed [33]. In UM, only two studies to date have shown that the level of LDH and the volume of liver metastases appear as independent prognostic variables [19,30]. Interestingly, another recent publication, studying the survival of patients with metastatic cutaneous melanoma treated with Pembrolizumab, shows that the tumor size of metastases remains an independent prognostic factor in multivariate analysis among other variables including the LDH level [34]. In that study, the tumor burden was measured on CT as the sum of the uni-dimensional measurement of 10 lesions per patient. Given the prognostic importance of hepatic tumor burden, it may be useful to perform a precise 3D measurement of all metastatic lesions. Nevertheless, these measures are difficult to achieve in current clinical practice. In addition, the particular presentation of LMUM, which often associates supracentimetric lesions with multiple small lesions less than 5 mm, makes this 3D measurement even more difficult. This is why for our study we decided to associate the surface of the largest lesion and the number of metastases.

Concerning the genomic status of UM, this variable was incomplete (97/224 patients) because at our center this investigation has only been conducted since 2007 and could be performed only by transscleral biopsy on large UMs or when the patient was enucleated. While it is well known that the genomic characteristics of UM are a major factor in metastatic risk for patients, their prognostic value once the patient becomes metastatic remains to be established. Interestingly, miliary disease found on MRI was not included in the selected variables. In our first retrospective study, miliary disease was a poor prognostic variable correlated with the number of metastatic lesions, but it was a peroperative description [9]. It is not surprising that it was not selected here, given that miliary disease is under evaluated by MRI. This does not call into question our previous results [20].

### 3.1. Nomogram Interests

This nomogram is of clinical interest because the variables selected by the multivariate analysis are easily accessible, provided that there is a liver MRI at the time of discovery of metastases. It can therefore be an aid to the care of patients in this particular context where the effectiveness of treatments is still limited. This nomogram is particularly discriminating for prolonged survival at 24 months. Our average Harrell C-Index value on 400 bootstrap samples was 0.71. In the Valpione study, the Harrell C indices were 0.75 for the primary population and 0.80 for the external cohort [19]. When we compared Valpione’s study to ours, we found nearly the same prognostic factors that integrated the nomogram [19]. The only difference was the PS status, which was not retained in our nomogram. This difference could be explained by the rate of a PS of 0–1 of 96% in our cohort compared with 90% in the primary cohort and 70% in the validation cohort in Valpione’s study [19]. The other selected variables were identical. The main difference was how we evaluated the percentage of hepatic invasion. In our study, we assumed that the total number of hepatic metastases associated with the surface of the largest lesion was a measure quite comparable to the precise calculation of the percentage of hepatic invasion, which is a time-consuming task. We thus believe that the bi-dimensional measurement of lesions, in line with the international WHO criteria, is a simpler and more reproducible method than true measurement of tri-dimensional liver tumor burden [32]. Liver MRI has already demonstrated its added value compared to CT in the evaluation of metastatic disease in other types of cancer and we expect that the same holds true for LMUM, although no comparative study has been published so far. Unlike the Valpione’s study, where only a minority of patients were explored by MRI in only one of the two study centers, all the patients in our study were explored by MRI [19]. The examinations were performed with a consistent routine protocol according to the time period, which probably contributed to the quality of the collected imaging data.

### 3.2. Nomogram Limitations

This study has limitations. First, due to liver US screening, which may lead to false negative cases, some of our patients may not have been detected at the onset of their metastatic disease. This limit is, however, common to other studies that use an imaging screening method with imperfect detection sensitivity. Second, the lower number of patients with survival less than 0.5 to 12 months explains the loss of accuracy of our nomogram in predicting shorter survival. Third, the nomogram was based on data collected at a single institution. Fourth, the study population enrolled for the establishment of the nomogram consisted of patients selected by liver MRI performed at a single institution over a long period of time. During this period, the technological evolutions in MRI and especially the parameters conditioning the spatial resolution could have influenced our results. However, this situation is difficult to avoid in retrospective studies of a rare disease. Therefore, this predictive model could be applied only for patients who had the same imaging modality. Fifth, a validation study using an external cohort is required to confirm the usefulness of this nomogram. Therefore, we plan to validate our nomogram on a test patient cohort presenting the same type of management at centers using MRI as a means of hepatic screening imaging. Sixth, in this retrospective study, we could not evaluate the prognostic value of new circulating biomarkers, such as CTC or ctDNA. Among the available data, two studies published by our team showed that there was a good correlation between the values of these biomarkers and the intrahepatic tumor burden volume calculated by CT or MRI [35,36]. However, the currently available results were too preliminary to be incorporated into the construction of a nomogram. Seventh, only a few studies have compared the genomic profiles of the primary tumor and metastases, and only one study showed the negative impact of GNA11 mutation on the disease specific survival and the overall survival of the 30 metastatic patients studied [37,38,39]. Due to the lack of genomic data on the metastases in this cohort, we could not study this parameter or its impact on survival, though we hope to do so in the near future.

## 4. Materials and Methods

### 4.1. Study Design and Participants

Among 6392 patients with UM registered since 1980 at Institut Curie, 3636 patients were selected, corresponding to the period 2002–2013. In 2002, the Picture Archiving Communication System (PACS) was installed, in which all the imaging data concerning the patients treated at our institution were archived. The cohort was truncated in 2013 to create a follow-up of more than two years for the last patients included. Throughout the period, 725 (20%) patients developed metastatic disease. Following international recommendations during the period of our study, all patients treated for UM were screened by liver US every 6 months. In case of abnormal or doubtful ultrasound results, a liver MRI was performed to establish the diagnosis and evaluate the importance of hepatic invasion. Among these patients, we selected patients who were treated at our institution for the full course of their treatments and who had a liver MRI performed in our center at the onset of their metastatic disease that was archived on the PACS and retrospectively analyzable. Two hundred and twenty four patients constituted the final selected cohort. This study was approved by the uveal melanoma clinical research staff and the Institut Curie Review Board (MU-06-2016) and was carried out according to the Declaration of Helsinki. Informed consent was waived because this was a retrospective case series that involved no diagnosis or therapeutic intervention.

The following continuous or categorical variables for each patient were collected—sex, age at diagnosis of UM, date of diagnosis of UM, clinical and genomic characteristics of UM, UM treatment, synchronous or non-synchronous LMUM, and disease-free interval from the treatment of UM to the diagnosis of LMUM. For metastatic disease, the following clinical and biological prognostic factors previously described in the literature were analyzed: Performance Status (PS) according to World Health Organisation classification, LDH rate, treatment of LMUM, the date of the last follow-up or death, and cause of death. Using MRI, we analyzed uni- or bilobar localization in the liver, number of hepatic segments involved, number of LMUM, surface area of the largest LMUM, and presence or absence of miliary disease [17,24,25,26].

### 4.2. Radiological Evaluation

MR imaging was performed on a 1.5 T clinical system [Siemens Healthcare]. In summary, the liver MRI protocol included axial 2D fat suppressed fast-spin echo (FSE) T2-weighted with respiratory trigger, breath hold axial 2D dual gradient-recalled echo (GRE) T1-weighted, and breath hold axial dynamic contrast-enhanced 3D GRE T1-weighted images before and after injection of an extracellular gadolinium contrast agent (arterial, portal, and delayed phase). A respiratory triggered echoplanar diffusion-weighted imaging (DWI) had not been routinely performed since 2004. The maximum b value was 600 s/mm2 between 2004 and 2009; and 800 s/mm2 between 2009 and 2013.

During the 2002–2009 period, slice thickness was 8 mm for 2D sequences and 5 mm for 3D sequences. During the 2009–2013 period, slice thickness was 6 mm for 2D and 3.5 mm for 3D sequences.

All MRIs were retrospectively reviewed by a single radiologist specialized in this disease, regardless of the clinical data. The morphological criteria used to diagnose metastases were published previously [20]. Lesions were counted per segment according to Couinaud’s classification [40]. The size of the largest lesion was measured in two dimensions [32]. For each examination, the presence or absence of miliary disease was noted. A miliary disease was defined as the presence of 3 or more metastatic lesions of less than 5 mm, whether subcapsular or intra-parenchymal. For each patient, extrahepatic disease was also assessed by the analysis of thoraco-abdomino-pelvic CT and bone scintigraphy.

### 4.3. Statistical Analysis

Quantitative variables are described as median with inter-quartile interval. The qualitative variables are described with the numbers of each class and their corresponding percentage. The criterion of interest was overall survival, which was defined as the time from the date of metastasis diagnosis to the occurrence of death from any cause. Living patients were censored on the date of the last follow-up. The survival curves were estimated according to the Kaplan–Meier method and compared using the log-rank test. Univariate and multivariate analyses were performed to investigate the prognostic factors for survival. For multivariate analysis, a Cox proportional hazard model was used. A stepwise procedure using bootstrapping and based on the Akaike Information Criterion (AIC) was performed to select variables. Continuous variables were categorized into clinically relevant classes. The final predictive model obtained was internally validated. Discrimination and calibration of the model were evaluated using bootstrapping to take into account the bias of optimism resulting from the fact that the same patients were used to establish the predictive model and to validate it. Discrimination of the model was evaluated via the Harrell C-Index. The calibration of the model was evaluated graphically with calibration curves at the three times of interest presented in the nomogram—6, 12, and 24 months. The nomogram was depicted with the confidence intervals of each variable, thus allowing an estimate of the confidence interval of survival. The analyses were all carried out using R software (R version 3.2.2, R Core Team (2018), Vienna, Austria).

## 5. Conclusions

We proposed a nomogram that is clinically simple to use, integrating MRI imaging data to improve the management of the patients. The four independent prognostic factors that make up this nomogram are easily collected—time to onset of metastasis, number of LMUM on MRI, surface area of the largest LMUM on MRI, and LDH value at the diagnosis of LMUM. This tool could allow discussing the therapeutic options with each patient more objectively to find one that appears most adapted to him/her.

## Figures and Tables

**Figure 1 cancers-11-00863-f001:**
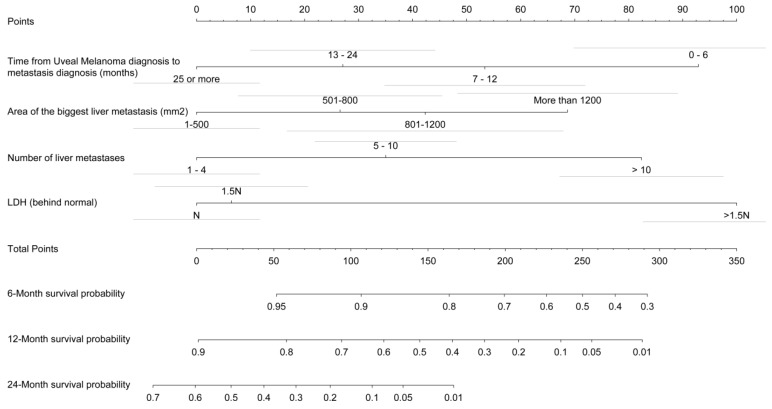
Nomogram of the final prognostic model. The sum of the prognostic factor points corresponds to the survival probability at 6, 12, and 24 months. On the nomogram, the first row corresponds to the row of points for the score of each variable. Next, we divided the prognostic factors into classes. The next line is the total score on which to sum the scores of the 4 variables in the nomogram. The last three lines correspond to the predicted survival at the 3 time points.

**Figure 2 cancers-11-00863-f002:**
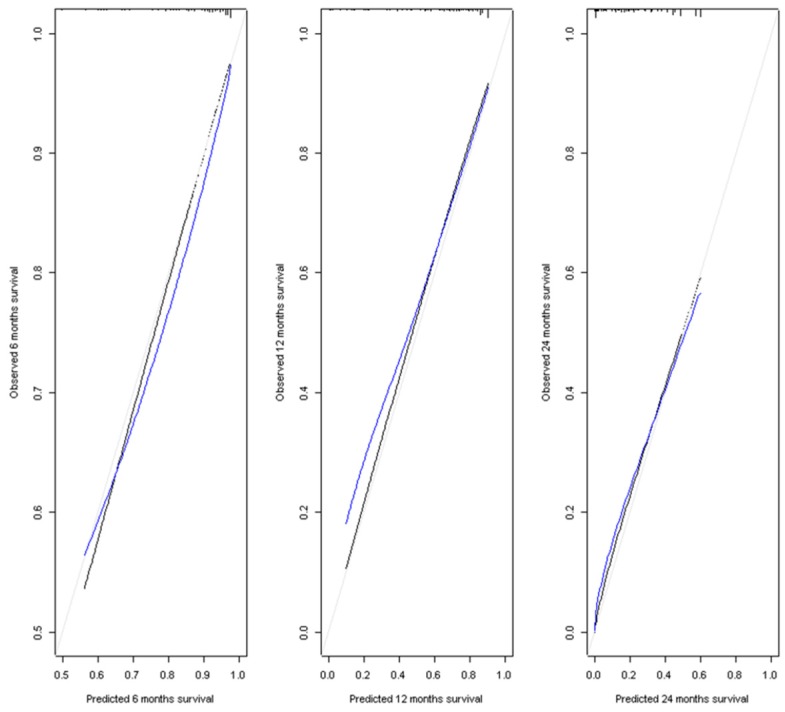
Calibration curves at 6, 12, and 24 months. Estimate of calibration accuracy was performed using adaptive spline regression. The line adjacent to the ideal line corresponds to the apparent predictive accuracy. The blue line corresponds to corrected estimates.

**Table 1 cancers-11-00863-t001:** Univariable analysis of factors associated with overall survival after diagnosis of liver metastases of uveal melanoma (LMUM).

Variables	Patients (*n* = 224)	%	HR (CI 0.95)	*p*
**Sex**				
*M*	113	50.4	1	
*F*	111	49.6	0.93 (0.69–1.26)	0.66
**Age at diagnosis of UM (year)**				
*Median value (IQR)*		57 (49–66)		
*≤60*	137	61.2	1	
*>60*	87	38.8	1.68 (1.23–2.30)	0.001
**Largest UM diameter (mm) (*n* = 221)**				
*Median value (IQR)*		17.1 (15–19.4)		
<18	128	57.9	1	
*≥18*	93	42.1	1.14	0.41
**Ciliary body involvement (*n* = 211)**				
*No*	127	60.2	1	
*Yes*	84	39.8	1.42 (1.03–1.96)	0.03
**Extra-scleral extension (*n* = 206)**				
*No*	191	92.7	1	
*Yes*	15	7.3	0.74 (0.40–1.37)	0.34
**Treatment of UM**				
*Enucleation*	95	42.4	1	
*Proton beam radiotherapy*	129	57.6	0.76 (0.56–1.03)	0.07
**Genomic analysis of UM (*n* = 97)**				
*High*	78	80.4	1	
*Intermediate*	18	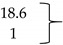	1.06 (0.54–2.08)	0.87
*Low*	1
**Disease-Free Interval between UM and LMUM**				
*Median value (IQR)*	26.1 (13.6–41.4)		
*0–6 months*	16	7.2	2.35 (1.35–4.1)	0.003
*6–12 months*	31	13.8	1.54 (0.97–2.44)	0.07
*12–24 months*	52	23.2	1.74 (1.19–2.53)	0.004
*>24 months*	125	55.8	1	
**Concomitant extrahepatic disease (*n* = 220)**				
*No*	199	90.5	1	
*Yes*	21	9.5	2.03 (1.31–3.16)	0.002
**First treatment of LMUM**				
*Surgery*	60	30.5	1	
*Systemic*	123	62.4	2.75 (1.80–4.2)	<0.001
*Best supportive care*	14	7.1	2.73 (1.43–5.24)	<0.002
**PS**				
*0*	160	71.4	1	
*1*	54	24.1	
*2*	9	4	1.87 (0.95–3.67)	0.07
*3*	1	0.4
**LDH (*n* = 221)**				
*Median value (IQR)*		394 (305–495)		
*≤N*	108	48.9	1	
*>N–≤1.5N*	78	35.3	1.30 (0.93–1.83)	0.13
*>1.5N*	35	15.8	4.15 (2.71–6.33)	<0.001
**Localization of liver metastases**				
*Bilobar*	147	65.6	1	
*Right lobe*	65	29	0.66 (0.46–0.94)	0.02
*Left lobe*	12	5.4	0.62 (0.31–1.23)	0.16
**Number of liver segments involved (MRI)**				
*1 to 3*	119	53.1	1	
*4 to 6*	66	29.5	1.32 (0.94–1.86)	0.11
*7 to 8*	39	17.4	3.57 (2.34–5.44)	<0.001
**LMUM number (MRI)**				
*1 to 4*	109	48.7	1	
*5 to 10*	57	25.4	1.27 (0.87–1.83)	0.22
*>10*	58	25.9	2.89 (2.0–4.18)	<0.001
**LMUM largest size (mm2) (MRI)**				
*Median value (IQR)*		361 (144–784)		
*[1–500]*	143	63.8	1	
*[501–800]*	27	12.1	1.17 (0.74–1.86)	0.51
*[801–1200]*	20	8.9	2.56 (1.56–4.19)	<0.001
*[1201–*	34	15.2	3.28 (2.14–5.02)	<0.001
**Miliary (MRI)**				
*No*	107	47.8	1	
*Yes*	117	52.2	1.43 (1.06–1.94)	0.02

HR, hazard ratio; IQR, Interquartile Range; CI, Confidence Interval; PS, Performance Status (according to World Health Organization classification); UM, uveal melanoma; LMUM, liver metastases of uveal melanoma.

**Table 2 cancers-11-00863-t002:** Multivariable analysis of factors associated with overall survival after diagnosis of LMUM.

Variables	HR (CI 0.95)	*p*
**Disease Free Interval between UM and LMUM**		
*>24 months*	1	
*12–24 months*	1.43 (0.94–2.16)	0.09
*6–12 months*	2.02 (1.24–3.27)	0.004
*0–6 months*	3.39 (1.90–6.05)	<0.001
**LDH**		
*≤N*	1	
*>N–≤1.5N*	1.09 (0.75–1.57)	0.65
*>1.5N*	3.72 (2.30–6.00)	<0.001
**LMUM number (MRI)**		
*[1–4]*	1	
*[5–10]*	1.58 (1.07–2.33)	0.02
*>10*	2.95 (1.97–4.43)	<0.001
LMUM largest size (mm2) (MRI)		
*[1–500]*	1	
*[501–800]*	1.42 (0.88–2.28)	0.15
*[801–1200]*	1.74 (1.00–3.05)	0.05
*[1201–*	2.47 (1.53–3.98)	<0.001

HR, hazard ratio; CI, confidence interval.

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
