# Peer review of "Development of a Prognostic Nomogram for Liver Metastasis of Uveal Melanoma Patients Selected by Liver MRI"

_cancers, 2019, doi:10.3390/cancers11060863_

Round 1

Reviewer 1 Report

The manuscript develops a nomogram for the life expectancy of UM patients with liver metastases based on MR-images of the liver. Although such a nomogram could aid in the determination of the optimal treatment for individual patients, the following points would improve the manuscript for publication:

General:

- The methods & materials section should be positioned before the results, so the manuscript can be read in a continuous manner. 

- The manuscript needs linguistic improvement by a native speaker (e.g. line 54 “… showed that actually surgical complete resection”, the word actually doesn’t belong at this position in the sentence)

Introduction:

- The first paragraph is a more a summary of statements and therefore reads very staccato. It should be rewritten to read more easily and subdivided in paragraphs.

- line 70: “MRI is currently the gold standard”. This should be more nuanced, since in many hospitals ultrasound is the commonly performed imaging modality, which also shows from the fact that from the 725 patients with metastasis, only 224 have had an MRI (if I understood the methods part correctly)

Methods:

 - the total number of included patients is not mentioned (it is in the results, but in the current form it is hard understand what the final study group consists of).

- more details about the MRI protocol should be given (type of scans, resolution). Some of these elements are probably mentioned in the references, but the main characteristics should be mentioned in this manuscript as well, so it can be read on its own.

- Given that the patients have been scanned over quite a long period of time, I assume that there are differences in the MRI protocol. Does this influence the radiological scoring, since an increase in resolution could make smaller lesions more easily detectable.

- line 221: why are the lesion dimensions only measured in 2D, and not in 3D since a 2D measurement could result in an underestimation of the tumor dimension. 

Discussion:

- It is surprising that two of the main factors in the univariate analysis (age at diagnosis and ciliary body involvement) do not play a role in multivariate analysis. This should be more elaborately discussed

- section 3.1 compares the MRI-based nomogram to the US/CT-based nomogram of Valpione. This discussion should be more elaborate and include the value of MRI (since apparently the same results are obtained as with US) and what this study adds.

Minor suggestions:

Lines 37-39: please mention which of the factors result in an increased risk of metastases

Author Response

The manuscript develops a nomogram for the life expectancy of UM patients with liver metastases based on MR-images of the liver. Although such a nomogram could aid in the determination of the optimal treatment for individual patients, the following points would improve the manuscript for publication:

General:

- The methods & materials section should be positioned before the results, so the manuscript can be read in a continuous manner.

Answer: We respected the journal’s instructions to the authors

- The manuscript needs linguistic improvement by a native speaker (e.g. line 54 “… showed that actually surgical complete resection”, the word actually doesn’t belong at this position in the sentence)

 Answer: We have revised the writing of the entire manuscript.

Introduction:

- The first paragraph is a more a summary of statements and therefore reads very staccato. It should be rewritten to read more easily and subdivided in paragraphs.

Answer: We have rewritten the introduction to be easier to read. We sub-divided in paragraphs.

- line 70: “MRI is currently the gold standard”. This should be more nuanced, since in many hospitals ultrasound is the commonly performed imaging modality, which also shows from the fact that from the 725 patients with metastasis, only 224 have had an MRI (if I understood the methods part correctly)

Answer: We also perform liver US for our patients. However, when the results of liver US show metastases or is doubtful, we complete the assessment in imaging by a thoraco-abominopelvic CT scan and a liver MRI when it seems useful. At this stage, it seems important to have an objective cross-sectional imaging that can be analyzed in a collegiate way to help the therapeutic decision.
The number of patients for whom we had an MRI performed in our institution and archived on the PACS is only 224 because many of these patients had only a CT-scan and others had an MRI performed in other imaging centers that was not archived in our PACS.
Although there is no publication on the subject, our experience is that the importance of liver metastatic disease is better assessed by MRI than by CT or US. This finding is consistent with the evaluation of liver metastases of other types of cancer (colon cancer, endocrine tumor, pancreatic cancer ...) for which comparative studies are available.

We agree with your remark regarding line 70 and have modified "Indeed, MRI is currently the gold standard for intrahepatic metastatic invasion in this type of metastases; MRI has been performed for a minority of patients in the validation cohort of this nomogram [20-22]." by "According to our experience, MRI is an imaging method for objectively and retrospectively assessing the importance of hepatic involvement in UM, like in other types of cancer [20-22].”

Methods:

 - the total number of included patients is not mentioned (it is in the results, but in the current form it is hard understand what the final study group consists of).

Answer: We added the number of patients of the final cohort in material and method: "224 patients thus constitute the selected cohort"

- more details about the MRI protocol should be given (type of scans, resolution). Some of these elements are probably mentioned in the references, but the main characteristics should be mentioned in this manuscript as well, so it can be read on its own.

Answer: We have made the following changes

In paragraph 4.2 Radiological evaluation: " MR imaging was performed on a 1.5 T clinical systems [Siemens Healthcare]. In summary, the liver MRI protocol included axial 2D fat suppressed fast-spin echo (FSE) T2-weighted with respiratory trigger, breath hold axial 2D dual gradient-recalled echo (GRE) T1-weighted and breath hold axial dynamic contrast-enhanced 3D GRE T1-weighted images before and after injection of an extracellular gadolinium contrast agent (arterial, portal and delayed phase). A respiratory triggered echoplanar diffusion-weighted imaging (DWI) was not routinely performed since 2004. The maximum b value was 600 s/mm2 between 2004 and 2009 and 800 s/mm2 between 2009 and 2013.”

- Given that the patients have been scanned over quite a long period of time, I assume that there are differences in the MRI protocol. Does this influence the radiological scoring, since an increase in resolution could make smaller lesions more easily detectable.

Answer: We agree with your comment since the slice thickness, the intersection gap and the matrix resolution will influence the detection of the infracentimetric lesions. This is especially true for lesions of 5mm or less and corresponds to our published experience. Nevertheless, for statistical reasons, it seemed difficult for us to separate our cohort into two distinct groups taking into account technological developments in MRI on the study period. However we have indicated in the text the following clarifications in the same paragraph: "During the 2002-2009 period, slice thickness was 8mm for 2D sequences and 5mm for 3D sequences. During the 2009-2013 period, slice thickness was 6mm for 2D and 3.5mm for 3D sequences. "

We also added in the paragraph 3.2 Nomogram limitations: “Third, the study population enrolled for the establishment of the nomogram consisted of patients selected by liver MRI performed at a single institution over a long period of time. During this period, the technological evolutions in MRI and especially the parameters conditioning the spatial resolution could have influenced our results. However, this situation is difficult to avoid in retrospective studies of a rare disease.”

- line 221: why are the lesion dimensions only measured in 2D, and not in 3D since a 2D measurement could result in an underestimation of the tumor dimension. 

Answer: We decided to use a simple, easy to use and reproducible measurement method. For the evaluation of tumors, the current international recommendations use uni-dimensional (RECIST 1.1) or two-dimensional measurements (WHO or Cheson criteria) and we have therefore followed these recommendations. A three-dimensional measurement of the lesions can be performed in several ways. We can segment manually or with software tumor lesions to obtain an exact measurement of the volume (as suggested in the Valpione paper). This method seems to us far too complicated to implement in the usual clinical practice. We can also, what we think you suggest, measure the tumor in three axes and calculate its volume approximately according to a predefined model (sphere,ellipsoid). This implies for us that we have an MRI 2D sequence in another plane (ex. coronal or sagittal) than the axial plane or a 3D sequence in high resolution allowing quality image reconstructions. Since our routine MRI protocol only includes axial sequences, we preferred to choose a two-dimensional measurement passing through the largest axis of the tumor in the axial plane like in international recommendations.

Discussion:

- It is surprising that two of the main factors in the univariate analysis (age at diagnosis and ciliary body involvement) do not play a role in multivariate analysis. This should be more elaborately discussed

Answer: For the nomogram it was necessary to select variables, those that seemed to have the greatest impact on survival. After selection, we obtained the four variables that are in the nomogram. In the paragraph discussion we have added: “Notably, age and ciliary body involvement are not retained in the final model because the information they convey and their effect on survival is at best minimal after adjusting for all variables. One of the hypotheses is that the ciliary body involvement, which is a known prognostic factor for the onset of metastases, has no effect once the patient is metastatic. It is possible that the characteristics of the initial tumor do not affect survival as soon as metastases have appeared, or that their effect is embedded within the DFI variable if they have contributed to metastases occurring faster”.

 - section 3.1 compares the MRI-based nomogram to the US/CT-based nomogram of Valpione. This discussion should be more elaborate and include the value of MRI (since apparently the same results are obtained as with US) and what this study adds.

Answer: Patient management in the Valpione study is similar to ours. After the discovery of the ocular tumor, patients are followed by ultrasound. Once the diagnosis of metastasis is established or suspected on liver US, complementary staging by CT or MRI is performed. In the publication of Valpione, it is indicated that the extension of metastases is quantified in 3D on CT or MRI without describing in detail the method employed: is it an accurate measurement of tumor volume or approximation? We believe that the two-dimensional measurement of the largest lesion associated with the number of metastases is much simpler to perform in clinical practice. This aspect is already emphasized in the discussion in the paragraph 3.1 “nomogram interests”.
In the Valpione study, the evaluation of tumor invasion of the liver is carried out for the majority of patients in both centers with CT and for a minority of patients (the number is not specified) of the USA center on liver MRI. No technical details concerning the MRI are specified in the publication with a study period extending between 2000 and 2013. The added value of our study is that all the patients benefited from a liver MRI performed in our center. In our opinion, hepatic MRI allows for better evaluation of hepatic tumor invasion in this disease as in other types of cancer.

We have added in the paragraph 3.1 Nomogram interests the following elements: "Liver MRI has already demonstrated its added value compared to CT in the evaluation of metastatic disease in other types of cancer and we expect that the same holds true for LMUM, although no comparative study has been published so far. Unlike the Valpione’s study, where only a minority of patients were explored by MRI in only one of the two study centers, all the patients in our study were explored by MRI. The examinations were performed with a consistent routine protocol according to the time period, which probably contributed to the quality of the collected imaging data.”

Minor suggestions:

Lines 37-39: please mention which of the factors result in an increased risk of metastases

Answer: we added in the text line 37 the two main clinical factors: “characteristics of UM (largest basal diameter, maximum tumor thickness) and on genomic…”

Reviewer 2 Report

Authors established a nomogram to choose a treatment in patient with metastatic uveal melanoma. The manuscript is really interesting with a good review of the literature. I just suggest to the Authors to insert also morphological prognostic factors of Uveal melanoma such as epitheliod cell type and vascular loops

Author Response

Authors established a nomogram to choose a treatment in patient with metastatic uveal melanoma. The manuscript is really interesting with a good review of the literature. I just suggest to the Authors to insert also morphological prognostic factors of Uveal melanoma such as epitheliod cell type and vascular loops.

Answer: We agree with your observation since these two factors are known as metastatic risk factors for primary ocular tumor. Nevertheless, only 42% of the patients in our series were enucleated, the others having been treated with proton beam radiotherapy. Therefore, pathology data are missing for the majority of patients, resulting in a lot of missing data for statistical analysis.

Reviewer 3 Report

Dear authors,

interesting work, however, the parameters of the proposed nomogram are not really surprizing. It is clear and established that the more metastases you have and the higher the LDH (as a marker for tumor load) the worse the prognosis is. In addition, it is known for a long time already, that the faster metastases occur after primary the more aggressive a disease is with a worse clinical outcome. Somewhat new is the measurement of a large metastases and the nomogram might be useful to calculate survival probabilies.

Major remarks:

1. In the introduction you write „The results of overall survival for these locally treated patients showed that actually surgical complete resection (R0) of metastases remains the most effective therapeutic method when it is possible , with a median overall survival of months. It corresponds to a survival gain of approximately 1 year compared with medical treatments [9, 10, 16].“ You cannot say that, as there is no randomized trial which shows that patients with oligometastases to the liver who do not get operated are worse in their clinical outcome. Long survival in these patients might just correlate with a slowly growing melanoma. I had a patient who survived with several systemic treatments in the metastasized stage for 10 years and I have no clue if the treatments helped him or if he was just a very slow progressor…

And you continue with „However, the surgical treatment is only possible in a quarter of patients because of the spread of metastatic disease in the liver, making it impossible in this case to obtain a complete resection of all metastases, which is the condition for gaining survival.“ ¼?! That’s very surprizing to me and far from my experience. It is more single patients….

And again „On the other hand, there is sometimes a significant benefit for some patients with a very long survival, although it is not possible to clearly identify the factors explaining this prolongation of survival [17, 18].“ You cannot state that they benefitted from surgery…

 ==> I would leave all this surgical part completely out the paper. This is not in the focus of your work, the introduction should be on what is known as prognostic markers for survival including data on LDH (cutaneous and uveal if there is), tumor load, etc

2. In addition you state in the introduction: „In this context, it seems useful to propose, based on the analysis of retrospective data, a nomogram to assist clinicians in selecting patients to offer them a treatment adapted to their life expectancy.“ A clinician would not choose the treatment based on the life expectancy but on the general health and the tumor load, e.g. patients with a low tumor load might have some time to be able to try PD-1 directed treatments before going into liver directed treatments. Systemic chemotherapies are only done if there are metastase outside the liver…..

3. Introduction „Only one study of this type has been published to date [19].“ And the following is discussion. I really miss in the introduction an introduction into prognostic markers such as LDH etc, please show for cutaneous melanoma and for uveal where you have.

4. Results: you do not explain your nomogram very well. If I get it right the worst 6-month survival probability is 30%?! With a survival probability of 30% for 6 months I would offer any treatment to a patient with the wish for treatment, no? I do not understand then how your nomogram would help. I really would suggest to rewrite your paper and focus on prognostic factors that you evaluated in a reasonable number of UM patients. Just correlate them with survival and develop the nomogram but do not suggest to use it for treatment decisions. That’s depending on so much more….

5. Discussion „Because these four factors can be assessed before the choice of treatment of LMUM, it is possible to identify patients who will benefit most from the treatment according to the overall survival at 6, 12 and 24 months.“ Please be very careful with such sentences, you cannot say that.

6. Conclusions: please conclude on the factors that you found to predict survival time and that the nomogram might help to talk with the patient. I would suggest to just give one final sentence as an outlook that the nomogram might be useful to guide treatment decisions also.

Minor remarks:

Table 1: „ intermediate“ mispelling

Table 1: the space for the HR should be wider that it fits in one line

Table 1: shouldn’t there be 3 lines for „genomic analysis“ HR?

Table 1: „first treatment of uveal melanoma“, „number of liver segments involved“ and others are out of format

Table 1: „LDH“ should be on the next page

Results, multivariate analysis: suddenly you say „MU“ instead of „UM“

Table 2: heading should be on next page

Table 2: the brackets are wrong in LMUM largest size

Table 2: misspelling in „confidence interval“

2.3.:  Is it predictive or prognostic?

2.3.: „On the nomogram, the first row corresponds to the row of points for the score of each variable. Next, we divided the prognostic factors into classes. The next line is the total score on which to sum the scores of the 4 variables in the nomogram. The last three lines  correspond to the predicted survival at the 3 time points. The bars above the different classes of the variables correspond to the confidence intervals. In terms of discrimination, the average Harrell C-Index value on 400 bootstrap samples was 0.71 (0.95 CI).“ That is a subheading of a figure, but not a results part

2.3. „The lower number of patients with survival less than 0.5 to 12 months explains this loss of accuracy in predicting shorter survival.“ That’s discussion

Figure 1: please check the brackets, wrong direction in some

Figure 1: „number of liver metastasis“ è should be plural

Author Response

Dear authors,

interesting work, however, the parameters of the proposed nomogram are not really surprizing. It is clear and established that the more metastases you have and the higher the LDH (as a marker for tumor load) the worse the prognosis is. In addition, it is known for a long time already, that the faster metastases occur after primary the more aggressive a disease is with a worse clinical outcome. Somewhat new is the measurement of a large metastases and the nomogram might be useful to calculate survival probabilies.

Major remarks:

In the introduction you write „The results of overall survival for these locally treated patients showed that actually surgical complete resection (R0) of metastases remains the most effective therapeutic method when it is possible , with a median overall survival of months. It corresponds to a survival gain of approximately 1 year compared with medical treatments [9, 10, 16].“ You cannot say that, as there is no randomized trial which shows that patients with oligometastases to the liver who do not get operated are worse in their clinical outcome. Long survival in these patients might just correlate with a slowly growing melanoma. I had a patient who survived with several systemic treatments in the metastasized stage for 10 years and I have no clue if the treatments helped him or if he was just a very slow progressor…

Answers : We agree with the reviewers that randomized trials are missing so far and thus we removed: “The results of overall survival for these locally treated patients showed that actually surgical complete resection (R0) of metastases remains the most effective therapeutic method when it is possible , with a median overall survival of months. It corresponds to a survival gain of approximately 1 year compared with medical treatments [9, 10, 16].” We agree with the reviewer to say that there are indeed some patients who are long survivors regardless of the type of treatment proposed (systemic or loco-regional). We now added this into the manuscript (introduction).

And you continue with „However, the surgical treatment is only possible in a quarter of patients because of the spread of metastatic disease in the liver, making it impossible in this case to obtain a complete resection of all metastases, which is the condition for gaining survival.“ ¼?! That’s very surprizing to me and far from my experience. It is more single patients….

Answer: The percentage of oligometastatic patients depends on the applied strategy for detection by imaging. Since many years, MRI allows us to detect patients with low metastatic load to whom we readily propose loco-regional treatments, among which surgery. We agree that this is not standard care in many centres treating uveal melanoma and therefore we removed the percentage from the text.

And again „On the other hand, there is sometimes a significant benefit for some patients with a very long survival, although it is not possible to clearly identify the factors explaining this prolongation of survival [17, 18].“ You cannot state that they benefitted from surgery…

Answer : this is not what we wanted to say, we simply wanted to underline, as the reviewer already said, that there were long survivors regardless of the treatment (systemic or loco regional), in accordance with the 2 references: (Hsueh, E.C.et al. Prolonged survival after complete resection of metastases from intraocular melanoma. Cancer 2004, 100, 122-9; Buzzacco, D.M. et al. Long-term survivors with metastatic uveal melanoma. Open Ophtal. J. 2012, 6, 49-53). We replaced this in the text by “Nevertheless, there is a population of long-term survivors, who have slowly progressive metastases and for which any treatment results in a stable disease. It is currently not clear whether this is due to the action of the proposed treatment or a naturally quiescent disease [17, 18].”

 ==> I would leave all this surgical part completely out the paper.

Answer : we removed all this surgical part :The results of overall survival for these locally treated patients showed that actually surgical complete resection (R0) of metastases remains the most effective therapeutic method when it is possible , with a median overall survival of 27 months.  It corresponds to a survival gain of approximately 1 year compared with medical treatments [9, 10, 16]. However, the surgical treatment is only possible in a quarter of patients because of the spread of metastatic disease in the liver, making it impossible in this case to obtain a complete resection of all metastases, which is the condition for gaining survival.  Nevertheless, these patients often have metastatic recurrence in the liver within a median time of 10 months, making questionable the use of local invasive therapies in the absence of effective systemic treatment to help this local treatment [11].”

Instead, we propose a new paragraph in the introduction about prognostic markers for survival including data on LDH (cutaneous and uveal if there is), tumor load, etc: “

Several retrospective studies have assessed the prognostic factors in patients with LMUM [23-28]. The main independent prognostic factors common to these different studies include clinical and biological factors, as well as the evaluation of liver tumor burden determined by different imaging methods. In three more recent retrospective studies [19,29,30], the previously described prognostic factors appear as the main factors determining patient survival: age, performance status, delay in metastasis (DFI), LDH level [19,30] or alkaline phosphatases [29] and the extent of hepatic invasion assessed on imaging. Hepatic invasion can be assessed either by measuring the largest diameter of the largest metastasis [29,30] or by measuring the percentage of liver invasion [19],

Among the previously mentioned prognostic factors, the measurement of the importance of hepatic invasion with imaging is the most variable and the least codified. The results may depend on the imaging method used (US, CT, MRI) and the method of measuring tumor invasion (number of metastases evaluated, one-dimensional (1D), two-dimensional (2D) or three-dimensional (3D) measurement using either geometric models or a precise manual segmentation of the lesions). If the international rules for the evaluation of tumors in 1D (RECIST 1.1) [31] or in 2D (WHO) [32] are well established, the overall assessment of tumor burden by 3D methods remains to be established. Given currently available data for metastatic uveal melanoma, the imaging mode used and the method of assessing tumor burden on imaging remains to be standardized. From this point of view, MRI has appeared for many years as the most efficient imaging method for evaluating the spread of liver metastatic disease in several types of cancer [21,22]. For many years we have been performing liver MRI in some of our patients suspected of metastatic evolution of their primary ocular tumor. We could therefore retrospectively analyze the MRI data archived in our center since 2002.

2. In addition you state in the introduction: „In this context, it seems useful to propose, based on the analysis of retrospective data, a nomogram to assist clinicians in selecting patients to offer them a treatment adapted to their life expectancy.“ A clinician would not choose the treatment based on the life expectancy but on the general health and the tumor load, e.g. patients with a low tumor load might have some time to be able to try PD-1 directed treatments before going into liver directed treatments. Systemic chemotherapies are only done if there are metastase outside the liver…..

Answer : The nomogram has been designed to help the clinician make a treatment decision. It does not seek to substitute itself for it and it does not exclude the other parameters of the decision like the general health of the patient, the tumor load as well as the motivation of the patient for the treatment. We emphasize that among the parameters of the nomogram we have integrated the status performance and the tumor load described by the different parameters of the MRI.

 We added this sentence:

« Because clinicians lack a standard prognostic tool, it seems useful to propose a reproducible prognostic algorithm, based on the analysis of retrospective data, that could be integrated in clinical practice.”.and removed :” it seems useful to propose, based on the analysis of retrospective data, a nomogram to assist clinicians in selecting patients to offer them a treatment adapted to their life expectancy “

Regarding the choice of a first treatment by anti-PD1, we also have this attitude in our institution but the current data available in the literature show disappointing results. We recently published a case of response in a hyper-mutated patient (Rodrigues M et al. Outlier response to anti-PD1 in uveal melanoma reveals germline MBD4 mutations in hypermutated tumors. Nat Commun. 2018 May 14;9:1866.) which is an exceptional situation in this disease known to have one of the lowest mutation rates among different tumors types.

3. Introduction „Only one study of this type has been published to date [19].“ And the following is discussion. I really miss in the introduction an introduction into prognostic markers such as LDH etc, please show for cutaneous melanoma and for uveal where you have.

Answer: we put a paragraph on the choice of the prognostic markers in the introduction.

Several retrospective studies have assessed the prognostic factors in patients with LMUM [23-28]. The main independent prognostic factors common to these different studies include clinical and biological factors, as well as the evaluation of liver tumor burden determined by different imaging methods. In three more recent retrospective studies [19,29,30], the previously described prognostic factors appear as the main factors determining patient survival: age, performance status, delay in metastasis (DFI), LDH level [19,30] or alkaline phosphatases [29] and the extent of hepatic invasion assessed on imaging. Hepatic invasion can be assessed either by measuring the largest diameter of the largest metastasis [29,30] or by measuring the percentage of liver invasion [19],

4. Results: you do not explain your nomogram very well. If I get it right the worst 6-month survival probability is 30%?! With a survival probability of 30% for 6 months I would offer any treatment to a patient with the wish for treatment, no? I do not understand then how your nomogram would help. I really would suggest to rewrite your paper and focus on prognostic factors that you evaluated in a reasonable number of UM patients. Just correlate them with survival and develop the nomogram but do not suggest to use it for treatment decisions. That’s depending on so much more….

Answer : we consider that survival information obtained from the nomogram can help the clinician to take care of the patient, taking into account other usual parameters.
We therefore added a sentence concerning this remark in the introduction and in the discussion.

In the introduction : “Because clinicians lack a standard prognostic tool, it seems useful to propose a reproducible prognostic algorithm, based on the analysis of retrospective data, that could be integrated in clinical practice.” and in discussion (paragraph nomogram interest) “This nomogram is of clinical interest because the variables selected by the multivariate analysis are easily accessible, provided that there is a liver MRI at the time of discovery of metastases. It can therefore be an aid to the care of patients in this particular context where the effectiveness of treatments is still limited.”

5. Discussion „Because these four factors can be assessed before the choice of treatment of LMUM, it is possible to identify patients who will benefit most from the treatment according to the overall survival at 6, 12 and 24 months.“ Please be very careful with such sentences, you cannot say that.

Answer : we suppressed this sentence. 

6. Conclusions: please conclude on the factors that you found to predict survival time and that the nomogram might help to talk with the patient. I would suggest to just give one final sentence as an outlook that the nomogram might be useful to guide treatment decisions also.

Answer : we wrote a new conclusion based on the suggestion of the reviewer: “We proposed a nomogram that is simple to use clinically, integrating MRI imaging data to improve the management of the patients. The four independent prognostic factors that make up this nomogram are easily collected: time to onset of metastasis, number of LMUM on MRI, surface area of the largest LMUM on MRI, and LDH value at the diagnosis of LMUM. This tool could allow discussing more objectively with each patient the therapeutic options appearing most adapted to him/her.”

Minor remarks:

Table 1: „ intermediate“ misspelling.

Answer: We modified intermediate

Table 1: the space for the HR should be wider that it fits in one line. Answer: We modified

Table 1: shouldn’t there be 3 lines for „genomic analysis“ HR  Answer: there was only one patient with low genomic status, we pooled for the univariable analysis the intermediate and low risk patients all together. We put a bracket in table 1.

Table 1: „first treatment of uveal melanoma“, „number of liver segments involved“and others are out of format. Answer: we modified

Table 1: „LDH“ should be on the next page , Answer: we modified

Results, multivariate analysis: suddenly you say „MU“ instead of „UM“ Answer: we did corrections

Table 2: heading should be on next page , Answer: we did

Table 2: the brackets are wrong in LMUM largest size, Answer: we modified

Table 2: misspelling in „confidence interval , Answer : we corrected

2.3.:  Is it predictive or prognostic? Answer: we modified in “prognostic”

2.3.: „On the nomogram, the first row corresponds to the row of points for the score of each variable. Next, we divided the prognostic factors into classes. The next line is the total score on which to sum the scores of the 4 variables in the nomogram. The last three lines  correspond to the predicted survival at the 3 time points. The bars above the different classes of the variables correspond to the confidence intervals. In terms of discrimination, the average Harrell C-Index value on 400 bootstrap samples was 0.71 (0.95 CI).“ That is a subheading of a figure, but not a results part

Answer: We put it in a subheading of figure 1

2.3. „The lower number of patients with survival less than 0.5 to 12 months explains this loss of accuracy in predicting shorter survival.“ That’s discussion

Answer:  we put it in the discussion.” This study has limitations. First, the lower number of patients with survival less than 0.5 to 12 months explains the loss of accuracy of our nomogram in predicting shorter survival.”

Figure 1: please check the brackets, wrong direction in some

Answer: we corrected this.

Figure 1: „number of liver metastasis“ è should be plural

Answer: we did.

Round 2

Reviewer 1 Report

The authors have significantly improved the manuscript. 

The author's response, however, raised one significant point: the authors state " when the results of liver US show metastases or is doubtful, we complete the assessment in imaging by a thoraco-abominopelvic CT scan and a liver MRI when it seems useful." This sounds like a bias towards a specific group of patients, as other groups with patients (eg. those without metastases and/or a good US) are not included. This should at least be clearly mentioned and discussed in the methods and discussion section.

Other small points:

The reply of the authors regarding the 2D or 3D tumor measurements is sufficient, but I think it would be good to also include this limitation in the discussion.

line 99: "the most efficient imaging method". I think this still has to be proven, since to be efficient also things like the time and costs to make an MRI should be evaluated. I would therefor just call it "an effective imaging method". 

Author Response

The authors have significantly improved the manuscript.

The author's response, however, raised one significant point: the authors state " when the results of liver US show metastases or is doubtful, we complete the assessment in imaging by a thoraco-abominopelvic CT scan and a liver MRI when it seems useful." This sounds like a bias towards a specific group of patients, as other groups with patients (eg. those without metastases and/or a good US) are not included. This should at least be clearly mentioned and discussed in the methods and discussion section.

Response:
We apologize if our answer was not clearly formulated.  In practice, our patients are followed by liver ultrasonography every 6 months. When the ultrasound result is questionable or abnormal, we perform a liver MRI to confirm metastatic involvement and assess its importance. We also perform a thoraco-abdomino-pelvic CT 
and a bone scintigraphy to complete the metastatic extension assessment (see chapter 4.2 Radiological evaluation line 295-297).
We are aware that ultrasound may not detect all metastatic patients (false negative cases). Nevertheless, at the time of the study and in line with the international recommendations concerning this pathology, all our patients had a screening by ultrasound. There are currently very few published studies concerning liver MRI screening, including a study from our group (Piperno-Neumann S. et al., Prospective study of surveillance testing for metastasis in 100 high-risk uveal melanoma patients. Journal français d’ophthalmology 2015; 38: 526-534). We have recently opened an MRI screening protocol for patients at high clinical and / or genomic risk that is currently underway.
To answer your question directly, it is certainly true that any false negative patients of ultrasound screening are not taken into account in our study. It is reasonable to assume that these patients have an early metastatic disease that is difficult to detect and remains quiescent, because if their disease progresses, the metastases would probably have been detected by the ultrasound performed 6 months later. This population of patients with limited and quiescent liver disease is probably under-represented in our cohort.

We have made the following changes :
- In Materials and Methods, chapter 4.1 Study design and Participants line 264: "Following international recommendations during the period of our study, all patients treated for MU were screened by liver 
US every 6 months. In case of abnormal or doubtful ultrasound result, a liver MRI was performed to establish the diagnosis and evaluate the importance of hepatic invasion. "
-In discussion, chapter 3.2 Nomogram limitations line 236: "First, due to liver US screening, which may lead to false negatives cases, some of our patients may not have been detected at the onset of their metastatic disease. This limit is, however, common to other studies that use an imaging screening method with imperfect detection sensitivity. "

Other small points:

The reply of the authors regarding the 2D or 3D tumor measurements is sufficient, but I think it would be good to also include this limitation in the discussion.

Response :
We made the following changes.
In the chapter Discussion line 201: "Given the prognostic importance of hepatic tumor burden, it may be useful to perform a precise 3D measurement of all metastatic lesions. Nevertheless, these measures are difficult to achieve in current clinical practice. In addition, the particular presentation of LMUM, which often associates supra-centimetric lesions with multiple small lesions less than 5mm, makes this 3D measurement even more difficult. This is why we decided for our study to associate the surface of the largest lesion and the number of metastases. "

line 99: "the most efficient imaging method". I think this still has to be proven, since to be efficient also things like the time and costs to make an MRI should be evaluated. I would therefor just call it "an effective imaging method".

Response: we made the change.

Round 3

Reviewer 1 Report

The authors have successfully adressed the remaining points.